# Assessment of Tumor-Infiltrating Lymphocytes in Triple-Negative Breast Cancer: Interobserver Variability and Contributing Factors

**DOI:** 10.3390/diagnostics15192492

**Published:** 2025-09-30

**Authors:** Nurkhairul Bariyah Baharun, Mohamed Afiq Hidayat Zailani, Afzan Adam, Qiaoyi Xu, Muaatamarulain Mustangin, Reena Rahayu Md Zin

**Affiliations:** 1Department of Pathology, Faculty of Medicine, The National University of Malaysia, Kuala Lumpur 56000, Malaysia; p110546@siswa.ukm.edu.my or nurkhairun@unisel.edu.my (N.B.B.); afiqhidayat@ukm.edu.my (M.A.H.Z.); amar@hctm.ukm.edu.my (M.M.); 2Department of Medical Diagnostic, Faculty of Health Sciences, Universiti Selangor, Shah Alam 40000, Malaysia; 3Centre for Artificial Intelligence Technology (CAIT), Faculty of Information Science & Technology, The National University of Malaysia, Bangi 43600, Malaysia; afzan@ukm.edu.my (A.A.); p114345@siswa.ukm.edu.my (Q.X.)

**Keywords:** tumor-infiltrating lymphocytes (TILs), triple-negative breast cancer (TNBC), manual TIL scoring, interobserver variability, intraclass correlation, diagnostic reproducibility

## Abstract

**Background/Objectives:** Tumor-infiltrating lymphocytes (TILs) are emerging as a crucial prognostic biomarker in triple-negative breast cancer (TNBC). However, their clinical utility remains constrained by the subjectivity and interobserver variability of manual scoring, despite standardization efforts by the International TILs Working Group (TIL-WG). This study aimed to evaluate the interobserver agreement among pathologists in scoring stromal and intratumoral TILs from H&E-stained TNBC slides and to identify contributing histological factors. **Methods:** Two consultant pathologists at Hospital Canselor Tuanku Muhriz, Kuala Lumpur, independently assessed 64 TNBC cases using TIL-WG guidelines. Interobserver agreement was quantified using the intraclass correlation coefficient (ICC) and Cohen’s kappa coefficient. Cases with over 10% scoring discrepancies underwent review by a third pathologist, and a consensus discussion was held to explore the underlying confounders. **Results:** Our results showed moderate interobserver agreement for stromal TILs (ICC = 0.58) and strong agreement for intratumoral TILs (ICC = 0.71). Significant variability was attributed to three main confounding variables: heterogeneous TIL distribution, poorly defined tumor-stroma interface, and focal dense lymphoid infiltrates. **Conclusions:** These findings highlight the need for standardized TIL scoring protocols and suggest that validated AI-based tools may help mitigate observer variability in future TIL assessments.

## 1. Introduction

Triple-negative breast cancer (TNBC) is an aggressive immunogenic [1], subtype of breast cancer characterized by the absence of estrogen, progesterone, and human epidermal growth factor receptor 2 (HER-2) [2], presenting significant challenges in prognostication and treatment due to limited targeted therapies. Within the tumor microenvironment, tumor-infiltrating lymphocytes (TILs) have emerged as a valuable prognostic biomarker in TNBC [3,4,5,6,7] with high densities correlating with improved patient outcomes, including overall and disease-free survival. Therefore, accurate TIL assessment is crucial for understanding the immune landscape in TNBC. The International TILs Working Group (TIL-WG) has established standardized guidelines for evaluating stromal TILs (sTILs) by quantifying their proportion within tumor-associated stromal area while also outlining specific exclusion criteria [8]. The density of sTILs can be calculated using the following formula:
$$\mathrm{sTILs}\;\mathrm{Density}=\left(\frac{\mathrm{Area}\;\mathrm{of}\;\mathrm{sTILs}}{\mathrm{Area}\;\mathrm{of}\;\mathrm{tumor}\;-\;\mathrm{associated}\;\mathrm{stroma}}\right)\times 100\%$$


In contrast, the density of intratumoral TILs (iTILs) is represented by the percentage of the area occupied by TILs within the area of the tumor epithelium [9]. The following formula describes the calculation for the iTILs density:
$$\mathrm{iTILs}\;\mathrm{Density}=\left(\frac{\mathrm{Area}\;\mathrm{of}\;\mathrm{iTILs}}{\mathrm{Area}\;\mathrm{of}\;\mathrm{tumor}\;\mathrm{epithelium}}\right)\times 100\%$$


Areas of regressive hyalinization, necrosis, TILs around in situ carcinoma, typical structures, and previous needle biopsy sites should be excluded in the assessment [8]. Despite these standardized criteria and endorsements for clinical use, manual visual assessment by pathologists remains inherently subjective [8,10]. This often leads to variability in scoring thresholds and significant interobserver discrepancies. Such inconsistencies can undermine the reliability of pathology reports, potentially impacting clinical trial outcomes and patient therapeutic decisions [11,12,13]. Consequently, ensuring the reproducibility of TIL scoring is paramount for consistent clinical management. Interobserver agreement, commonly assessed using the intraclass correlation coefficient (ICC) for continuous data and Cohen’s kappa for categorical data, is vital for evaluating TIL assessment reproducibility across different observers [14,15,16,17,18].

Previous studies have reported a wide range of ICC values (0.50–0.933) [11,16,17,19] and Cohen’s kappa values (0.21 to 0.881) [11,16,18] highlighting considerable variability in agreement levels influenced by factors such as study design, observer experience, and methodology. This variability stems from a complex interplay of biological, technical, and human factors. Biological contributors include intra-tumor heterogeneity, characterized by uneven TIL distributions leading to inconsistent region of interest (ROI) selection [15,16] as well as ambiguous features like abundant tertiary lymphoid structures or a poorly defined tumor-stroma interface [19,20]. Technical factors such as suboptimal slide quality, inadequate tissue processing, or artifacts can obscure cellular details, which are critical for accurate lymphocyte identification [21]. Furthermore, human elements, including pathologists’ varying experience levels, personal interpretation styles, and familiarity with TIL scoring guidelines, can lead to scoring discrepancies, particularly in challenging or borderline cases [22,23].

The observed discrepancies in agreement underscore a critical need for enhanced standardization and additional training in TIL assessment methodologies to improve consistency and ensure quality control in histopathological evaluations. Therefore, this study aims to evaluate the interobserver agreement among pathologists when scoring both sTILs and iTILs in TNBC cases utilizing a standardized assessment criterion. Secondarily, this study seeks to identify specific histological features that contribute to these scoring discrepancies.

## 2. Materials and Methods

### 2.1. Sample Collection and Dataset Description

A retrospective study was conducted on 64 TNBC cases diagnosed at Hospital Canselor Tuanku Muhriz (HCTM), Kuala Lumpur, between January 2012 and December 2021. Inclusion criteria comprised TNBC cases with no prior history of malignancies within 5 years of diagnosis, verified availability of formalin-fixed paraffin-embedded (FFPE) tissue blocks, and sufficient material for hematoxylin and eosin (H&E) and immunohistochemistry (IHC) analysis. Exclusion criteria included inadequate tissue quality, technical artifacts, or failure to meet staining standards, which may jeopardize TIL assessment.

TNBC status was confirmed through IHC for estrogen receptor (ER), progesterone receptor (PR), and human epidermal growth factor receptor 2 (HER-2). ER and PR negativity were defined as <1% positive tumor nuclei, per American Society of Clinical Oncology/College of American Pathologists (ASCO/CAP) guidelines [24]. HER-2 negative status was defined by IHC scores of 0, 1+, and 2+, without HER-2 gene amplification confirmed by fluorescence in situ hybridization (FISH) or dual-color dual hapten in situ hybridization (DDISH) in cases where the HER-2 score was equivocal. Selected FFPE tissue blocks were sectioned at 4 µm for H&E and IHC staining.

### 2.2. Pathologist Selection and Manual TIL Assessment

The selected and quality-checked slides were scanned at 20× magnification using the Pannoramic DESK II DW slide scanner. Figure 1 illustrates the general workflow of pathologists for manual TIL assessment and annotation. Two board-certified pathologists (P1 and P2) with comparable clinical experience and expertise in breast pathology independently assessed TILs using digital whole slide images (WSI). Both were trained on the standardized TIL assessment criteria established by the TIL-WG. Before independent scoring, both pathologists participated in a 2 h calibration session using representative TNBC whole slide images annotated according to the TIL-WG recommendations. This session focused on aligning interpretations regarding sTILs vs. iTILs compartments and handling ill-defined tumor–stroma boundaries for TIL scoring. These reference cases helped introduce the observers to morphological patterns and minimize scoring discrepancies due to interpretive variability.

Each pathologist independently annotated ROIs and scored sTILs and iTILs as continuous variables (0% to 100%). The annotation and scoring process were guided by corresponding IHC, CD4+, and CD8+ markers to enhance the accuracy of lymphocyte identification and to minimize ambiguity in distinguishing TILs within the tumor microenvironment. Stroma TILs were quantified as the percentage of tumor-associated stroma occupied by mononuclear inflammatory cells, excluding areas of necrosis, hyalinization, and in situ components. Intratumoral TILs were assessed as the proportion of lymphocytes within tumor epithelial nests. TIL score was further categorized using five binary cut-off systems: ≤10%, ≤20%, ≤30%, ≤40%, and ≤50%. Low and high TIL classifications were defined relative to each threshold (Table 1), and for cases exhibiting more than 10% scoring discrepancy between P1 and P2, a third independent pathologist (P3) adjudicated the score. The final consensus score was determined by averaging the two most concordant scores among the three reviewers.

### 2.3. Interobserver Agreement and Consensus Review

Interobserver reliability was analyzed for both categorical and continuous TIL scores. Cohen’s kappa coefficient was applied to assess agreement for categorical TIL classifications across the five cut-off thresholds with interpretation based on Landis and Koch criteria [25]. Intraclass correlation coefficients (ICCs) using two-way random effects models assessed agreement and consistency for continuous TIL data, with interpretation based on the guidelines of Koo and Li [26]. ICC (agreement) evaluates both the correlation and absolute differences between pathologists’ scores, accounting for systematic differences in scoring (e.g., if one pathologist consistently gives higher scores than another). In contrast, ICC (consistency) assesses whether the scores are correlated regardless of absolute differences, meaning it allows for consistent shifts in scoring patterns between raters. Bland–Altman plots were generated to visualize agreement and identify any systematic biases. For discrepant cases reviewed by P3, pairwise kappa and ICC values were recalculated to evaluate whether consensus review improved interobserver agreement.

## 3. Results

### 3.1. Interobserver Agreement Using Continuous Scores

To better understand interobserver variability, we examined the distribution of sTIL and iTIL scores independently provided by each pathologist. The mean sTIL score assigned by pathologist 1 was 26.31% (SD: 21.77), while pathologist 2 reported an average of 30.58% (SD: 23.59). For iTILs, the mean scores were 10.0% (SD: 12.0) and 18.2% (SD: 20.83) for pathologist 1 and pathologist 2, respectively. These descriptive statistics reveal overlapping distributions; despite this, they also reflect interobserver differences, particularly more significant in iTIL scoring. The wider standard deviations indicate substantial variability across cases, which may contribute to the observed disagreement, particularly near the defined cut-off values.

Assessment of interobserver agreement for continuous TIL scoring between pathologists P1 and P2 showed moderate reliability for sTILs with ICC values of 0.57 (agreement) and 0.58 (consistency), respectively (95% CI, *p* < 0.001). In contrast, iTILs exhibited stronger reliability, with ICC of 0.70 (agreement) and 0.75 (consistency), indicating good reliability (Table 2). The ICC (consistency) values were slightly higher than ICC (agreement), suggesting that although the raters ranked the cases similarly, some systematic scoring differences were present, particularly in sTIL assessments.

### 3.2. Interobserver Agreement Using Categorical Scores

Cohen’s kappa (κ) coefficient was calculated for five TIL cut-off thresholds (10%, 20%, 30%, 40%, and 50%) to assess categorical agreement between the two pathologists. For sTILs, kappa values ranged from 0.13 to 0.40, with the highest agreement observed at the 10% (κ = 0.40), indicating a moderate level of interobserver agreement. Similarly, iTILs showed kappa values ranging from 0.25 to 0.48, with the 40% cut-off demonstrating the highest level of agreement (κ = 0.48). Agreement declined with increasing cut-off thresholds, suggesting greater subjectivity in identifying higher TIL densities (Table 3).

To visualize the discordance among pathologists based on the sTIL and iTIL density scores, Bland–Altman plots were drawn (Figure 2A,B). Bland–Altman plots were constructed to visualize scoring variability. For sTILs, the mean difference between observers was −4.26, with wide limits of agreement (−52.89 to +44.35), indicating moderate to high variability, particularly at higher densities. For iTILs, the mean difference was −8.22 with narrower limits of agreement (−37.98 to +20), reflecting relatively better consistency (Figure 2).

A heat map comparing TIL density scores across all cases further illustrated the variability, particularly in cases with intermediate and high TIL density. Notable discrepancies were evident through color shifts, especially in the yellow-to-red spectrum, denoting differences in lymphocyte density estimates between observers (Figure 3).

### 3.3. Interobserver Agreement for Discrepant Cases

Out of the 64 TNBC cases, 36 exhibited scoring discrepancies greater than 10% between P1 and P2. These cases were re-assessed by a third pathologist (P3). Consensus scoring significantly improved agreement metrics. For sTILs, ICC values increased to 0.70 (agreement) and 0.81 (consistency), while iTILs showed even higher values of 0.81 and 0.84, respectively (Table 4).

Pairwise Cohen’s kappa (κ) coefficients analysis among all three pathologists revealed fair to substantial agreement (Table 5). For sTILs, the highest agreement occurred at the 50% cut-off (average κ = 0.67). For iTILs, the agreement was most robust at the 50% threshold (average κ = 0.74), indicating substantial agreement.

## 4. Discussion

### 4.1. Interobserver Agreement and Consensus Review for Manual TIL Assessment

Accurate and reproducible assessment of TILs is essential for their use as prognostic biomarkers in TNBC. In this study, we evaluated interobserver variability among pathologists in scoring sTILs and iTILs using both continuous and categorical metrics. Our findings demonstrated moderate agreement for sTILs (ICC = 0.57–0.58, *p* < 0.001) and good agreement for iTILs (ICC = 0.70–0.75, *p* < 0.001), aligning with previous studies that report higher reproducibility for iTILs due to their more well-defined localization within tumor nests [13,27]. Minor differences between ICC values for agreement and consistency, particularly in sTIL scoring, suggest that while raters generally ranked cases similarly, some variability in absolute scoring remained, particularly in more subjective stromal regions. This reinforces the importance of standardized scoring protocols and consensus calibration to improve reproducibility in sTIL evaluation.

In contrast, the higher agreement seen for iTILs suggests that identifying lymphocytes within tumor cell nests is more straightforward and consistent across observers [18]. The reduced reproducibility in sTIL scoring likely reflects the interpretive challenges associated with assessing lymphocytes dispersed across variable stromal regions. This heterogeneity may lead to subjective variation in identifying representative fields, especially in areas with ambiguous tumor-stroma boundaries or dense focal infiltrates.

### 4.2. Agreement Across Categorical Cutoffs

When assessed using a categorical threshold, agreement levels decreased with increasing cut-off values. The highest kappa coefficients for sTILs and iTILs were observed at the 10% threshold, suggesting that pathologists can more reliably identify cases with low TIL density. Agreement declined notably at higher thresholds (≥30%), reflecting increased subjectivity in interpreting moderate to dense lymphocyte infiltration. These findings underscore the importance of standardizing cut-off selection, particularly when using TILs in clinical decision making or trial stratification.

### 4.3. Assessment of Scoring Agreement Using Bland–Altman Analysis

The Bland–Altman plot for sTILs demonstrates a relatively wide spread between the upper and lower limits of agreement, indicating moderate to high variability in pathologist scoring, particularly in cases with high TIL density. This finding is consistent with previous studies that have reported greater subjectivity when evaluating densely packed or heterogeneously distributed lymphocytic infiltrates [20]. In contrast, the Bland–Altman plot for iTILs displays narrower limits of agreement, suggesting better interobserver consistency. Nonetheless, several data points still fall outside these limits, especially in the mid-to-high TIL range. This may be attributed to difficulties in delineating tumor borders or accurately identifying lymphocytes within epithelial nests, which can be less distinct than stromal regions. Overall, both plots reinforce that while the level of agreement is generally acceptable, notable variability persists in cases with intermediate to high TIL infiltration, underscoring the inherent challenges of manual TIL assessment in complex histological contexts.

### 4.4. Heatmap Visualization of Scoring Discrepancies Across TIL Density Levels

Our heatmap analysis revealed notable variability in TIL scoring, particularly in cases with intermediate to high TILs, for both sTILs and iTILs. Pronounced color shifts between yellow, orange, and red across similar cases reflected inconsistencies in the density estimate between observers, highlighting the subjectivity inherent in manual evaluation. To address this, one promising strategy is the implementation of artificial intelligence (AI)-driven models for automated TIL (aTILs) quantification. A recent systematic review identified 27 studies employing such approaches in breast cancer, with the majority utilizing deep learning architectures, such as convolutional neural networks (CNNs) and fully convolutional networks (FCNs), for tasks like image segmentation and lymphocyte detection. These models were generally trained using pathologist-annotated ground truth datasets, with 58% of the studies reporting moderate to strong correlation (R = 0.6–0.98) between AI-generated outputs and manual TIL scores [28], underscoring their potential to enhance reproducibility and diagnostic precision.

As a summary, the notable interobserver variability demonstrated in Table 3, Table 4 and Table 5 and Figure 2 and Figure 3, despite the implementation of a calibration session, highlights the challenge of achieving reproducibility in manual TIL scoring. This variability was most significant in regions with moderate-to-high TIL densities, where the lack of a clear morphological cut-off may have contributed to greater interpretive subjectivity. Such discrepancies highlight the intrinsic limitations of visual-based scoring, which remains highly context-dependent and susceptible to individual bias, particularly in histologically complex or ambiguous regions, such as those with poorly defined tumor–stroma interfaces or clustered lymphoid infiltrates. These findings align with previous literature and collectively emphasize the urgent need for more objective, standardized frameworks, potentially supported by digital pathology and AI-based quantification tools, to minimize inconsistencies and enhance reproducibility in both clinical and research settings.

Collectively, evidence from prior studies suggests that concordance between manual TIL (mTIL) assessment and automated (aTIL) scoring ranges from weak to strong, primarily influenced by the differences in algorithm design, training data, and study methodology [4,5,20,29,30,31,32,33,34,35]. While these findings reinforce the growing potential of AI-based tools in pathology, they also highlight significant variability in performance. Notably, the automated models may struggle to replicate the nuanced judgment applied by experienced pathologists, particularly in histologically complex cases. This underscores the importance of ongoing validation, refinement, and clinical benchmarking of AI algorithms before their routine integration into diagnostic workflows can be justified.

### 4.5. Impact of Consensus Review on Scoring Consistency

Of the 64 cases assessed, 36 showed scoring discrepancies exceeding 10% between the two primary observers. Following adjudication by a third pathologist, interobserver agreement improved markedly, particularly for sTILs (ICC = 0.70–0.81) and iTILs (ICC = 0.81–0.84). This highlights the utility of structured consensus review in enhancing scoring reliability and reducing variability. Pairwise kappa analysis confirmed these improvements, with substantial agreement achieved in iTIL scoring at the 50% cut-off (κ = 1.00 between P1 and P3). Overall, these findings showed that cases that underwent consensus review showed improved interobserver agreement, supporting the value of multi-observer assessment in reducing subjectivity and enhancing consistency, particularly in cases with initial scoring discrepancies.

Consensus review methodologies have proven valuable in diagnostic settings involving subjective assessments and may serve as a quality control measure in pathology workflows, especially in multicenter trials or AI training datasets [36]. These improvements underscore the importance of incorporating consensus-based scoring into clinical and research workflows, particularly when addressing cases that exhibit significant interobserver variability.

As demonstrated in Table 3, Table 4 and Table 5, the level of interobserver agreement measured by Cohen’s kappa varied depending on the cut-off thresholds applied for TIL categorization. Notably, we observed that kappa values for sTIL assessment tended to be lower at intermediate cut-offs, particularly around the 30% cut-off, which approximates the mean sTIL value in our cohort. This reduction in agreement is attributable to increased classification discordance when both observers assign scores that fall close to the cut-off, resulting in small numerical differences leading to disagreement in categorical classification (e.g., high vs. low TILs). In contrast, more extreme cut-offs (e.g., 10% and 50%) demonstrated higher kappa values due to the score stratification, whereby the majority of cases fall uniformly into one category, thereby reducing the possibility of disagreement, even in the absence of true concordance. This phenomenon highlights a key limitation in interpreting kappa statistics, especially when cut-off-based classification is applied to continuous scoring data. These findings emphasize the importance of selecting cut-off points not only based on clinical validity but also with consideration of the score distribution to avoid skewed agreement metrics that could misrepresent the true extent of interobserver variability.

### 4.6. Contributing Factors to Interobserver Discrepancies

To better understand the factors causing disagreements in scoring, the discrepant cases were reviewed in detail. Discussions with the participating pathologist were undertaken to gather expert perspectives on the potential histological factors underlying the lack of agreement. Several potential contributors to interobserver discrepancies in TIL scoring have been identified. Among these, three key morphologic features were most consistently associated with discrepant cases: (1) heterogeneous distribution of TILs, (2) poorly defined tumor–stroma boundaries, and (3) focal dense lymphoid aggregates. Each of the 36 discrepant cases was reviewed and categorized accordingly. The frequency of each feature is summarized in Table 6. The majority of discrepant cases (32%) demonstrated heterogeneous TIL distribution, followed by poorly defined tumor–stroma boundaries (30%) and focal lymphoid aggregates (20%). Several cases exhibited more than one contributing factor, suggesting that combined histological complexity can exacerbate scoring variability. The findings are summarized in Table 6, providing a more precise visualization of the frequency and nature of each contributing factor across discrepant cases.

Other contributing factors to scoring discrepancies include the presence of necrosis and immune cell mimics, such as apoptotic bodies or reactive stromal cells, which may further interfere with accurate TIL identification and contribute to interobserver variability. Reactive plasma cells sometimes closely resembled tumor cells, leading to possible misinterpretation during assessment. Cases with extensive tumor necrosis made it challenging to distinguish between viable tumor tissue and the surrounding stroma, which in turn obscured the identification of infiltrating lymphocytes. These sources of error are inherently subjective and can mislead even experienced pathologists. In several cases, TILs were densely packed into small, focal areas, making it difficult to determine whether those regions accurately represented the overall immune response. This led to different observers choosing different regions for evaluation, which naturally contributed to scoring differences. These findings are consistent with previous reports indicating that tissue complexity, biological heterogeneity, and technical quality significantly influence TIL interpretation [19,20,37,38]. Such factors can lead to notable variability, even among experienced pathologists, particularly for ambiguous histological features.

Clinicopathological factors such as tumor size, nodal involvement, histological grade, and prior exposure to neoadjuvant chemotherapy may also influence interobserver variability in manual TIL scoring. These factors can significantly modulate the immune microenvironment, thereby affecting the density and spatial distribution of tumor-infiltrating lymphocytes. For instance, tumors with larger size, extensive necrosis, or treatment-induced regression patterns may present with heterogeneous immune cell infiltration, leading to interpretive challenges during manual assessment. Structured training programs focused on these morphologic pitfalls, along with consensus guidelines on region-of-interest (ROI) selection, could help reduce this variability and improve scoring consistency.

The absence of a widely accepted gold standard for TIL quantification remains a significant challenge in clinical and research applications. Although the TIL-WG has developed guidelines to enhance consistency in sTIL assessment, subjectivity persists due to interpretive variations, particularly in morphologically complex tissue regions. In this study, the observed interobserver variability highlights the inherent limitations of manual scoring approaches, which are influenced by factors such as heterogeneity in TIL distribution, ill-defined tumor–stroma boundaries, and focal dense lymphoid infiltrates. These challenges emphasize the need for developing a standardized quantification framework for TIL assessment. In parallel, there is growing support for the integration of artificial intelligence (AI)-based tools, which offer the potential to improve consistency and objectivity in TIL quantification, especially in histologically complex or borderline cases. Despite advances in automated detection and multi-target segmentation, the clinical adoption of AI in pathology remains limited. Barriers include interobserver variability in annotated datasets, complex tissue morphology, inconsistent labelling standards, and the limited transparency of AI decision-making processes [39,40]. These challenges highlight the need for robust, interpretable, and scalable AI models that can be seamlessly embedded into real-world pathology workflows. This is particularly critical in TNBC, where accurate TIL quantification is essential for prognostic classification and treatment planning.

Although this study did not directly evaluate AI, the findings provide a valuable benchmark for future studies that aim to compare manual scoring with automated approaches. Moving forward, combining expert review with AI-driven support systems could be key to improving the reliability of TIL assessment in both clinical and research settings.

### 4.7. Limitations and Future Directions

This study presents several novel contributions to the field of TIL assessment in TNBC, including a dual-compartment analysis of both stromal and intratumoral TILs, the integration of CD4+ and CD8+ IHC as visual aids to guide manual annotation on H&E-stained slides, and a descriptive, semi-quantitative examination of histopathological features underlying interobserver discrepancies. By incorporating these elements, our work offers a more comprehensive and biologically grounded evaluation of TILs in TNBC while also addressing critical sources of interpretive variability not sufficiently explored in previous interobserver studies.

This study has several limitations that warrant consideration. First, the sample size was relatively modest (*n* = 64), which may limit the generalizability of the findings to broader TNBC populations. Second, although the pathologists involved had comparable experience and were trained in standardized scoring protocols, inherent subjectivity in manual TIL assessment could still influence outcomes. Third, while interobserver variability was thoroughly examined, the study did not assess intra-observer consistency, which is also relevant for clinical reproducibility. Although IHC staining with CD4 and CD8 markers was used to assist pathologists in estimating TIL density during manual TIL scoring, the inclusion of another T-cell marker, such as CD3, could have offered a more comprehensive representation of total TIL infiltration. Additionally, although the potential of AI-assisted TIL scoring was discussed, no automated tools were directly evaluated in this study. This limits our ability to draw empirical conclusions about the comparative performance of manual versus automated approaches. Finally, the absence of clinical outcome correlation (e.g., survival or treatment response) restricts interpretation of the prognostic relevance of the observed scoring discrepancies.

While this study establishes the reliability and limitations of manual TIL scoring, future work should explore the integration of automated digital pathology tools for the quantification of TIL. Given the subjectivity and variability inherent to manual assessments, especially in morphologically complex regions, AI-based models offer an attractive pathway for reproducibility and scalability. Recent studies have demonstrated that deep learning approaches, CNNs and FCNs, can accurately segment lymphocytes and quantify TILs across large histological fields with minimal observer bias [20,33,34,35]. Tools such as QuPath, high-throughput analytics for learning and optimization (HALO) AI, and in-house trained pipelines have achieved moderate to strong correlation (r = 0.6–0.98) with pathologist-annotated ground truth. Incorporating such tools into our digitized TNBC slide dataset could provide valuable comparative insights into scoring consistency and highlight cases where AI either resolves or contributes to interobserver disagreement.

A potential analytical pipeline would involve running AI-based TIL quantification on the same set of whole-slide images and evaluating agreement metrics (e.g., ICC, Bland–Altman, and Spearman correlation) against consensus human scores. Additionally, discordant or outlier cases, particularly those resolved through adjudication by a third pathologist, could be re-evaluated to determine if AI models align more closely with consensus outcomes. This would serve to benchmark the practical utility of AI in reducing scoring bias and improving throughput in clinical settings. While the current study focused exclusively on assessing interobserver variability in manual TIL scoring using H&E-stained slides, we acknowledge the transformative potential of automated image analysis tools, such as ImageJ (https://imagej.net/software/imagej/ accessed on 3 February 2025), FIJI (https://imagej.net/software/fiji/ accessed on 3 February 2025), and QuPath (https://qupath.github.io/ accessed on 3 February 2025), in establishing reproducible reference standards. These platforms enable high-throughput and scalable TIL quantification, which could address subjectivity and enhance consistency across institutions. Ultimately, validating AI outputs against high-fidelity consensus scores may allow the development of hybrid models, where human oversight is retained for ambiguous regions, and AI handles bulk quantification. Future efforts should focus on building explainable AI frameworks that incorporate histological context, handle variable TIL distributions, and provide confidence scores to guide clinical decision making.

Another limitation of the present study is the lack of stratification based on clinicopathological parameters, such as tumor size, lymph node status, histological grade, and treatment history. These variables are known to influence the tumor immune landscape and may confound the interpretation of TIL density and distribution. Stratified evaluation of TIL scoring agreement based on clinical subgroups will provide a more detailed understanding of the contexts in which variability is most distinct and may inform more refined scoring strategies.

## 5. Conclusions

This study highlights the persistent interobserver variability in manual TIL assessment among pathologists, particularly for sTILs in TNBC cases. While moderate to substantial agreement was achieved, especially following consensus review, the findings reveal significant discrepancies in intermediate to high TIL density cases, driven by histological complexity and interpretive subjectivity even when standardized guidelines are followed. Such variability highlights the limitations of manual scoring even when standardized guidelines are applied. These findings reflect the intrinsic limitations of manual scoring approaches and reinforce the urgent need for more consistent, objective, and scalable methods of TIL quantification. Given these challenges, future efforts should focus on cross-validating AI outputs with expert consensus scores, incorporating clinical outcome data to refine cut-offs, and developing hybrid workflows that leverage both computational precision and pathologist oversight. By bridging manual expertise with automated tools, the field can move closer to establishing TILs as reliable, standardized biomarkers in personalized breast cancer management.

## Figures and Tables

**Figure 1 diagnostics-15-02492-f001:**
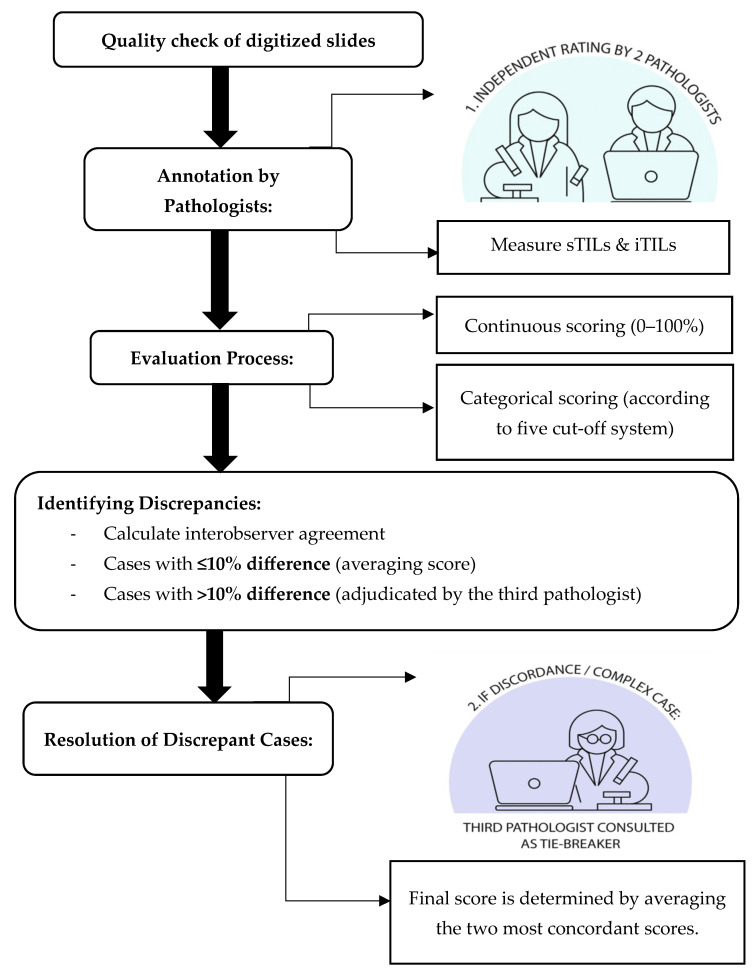
Workflow of manual TIL assessment and annotation by the pathologist.

**Figure 2 diagnostics-15-02492-f002:**
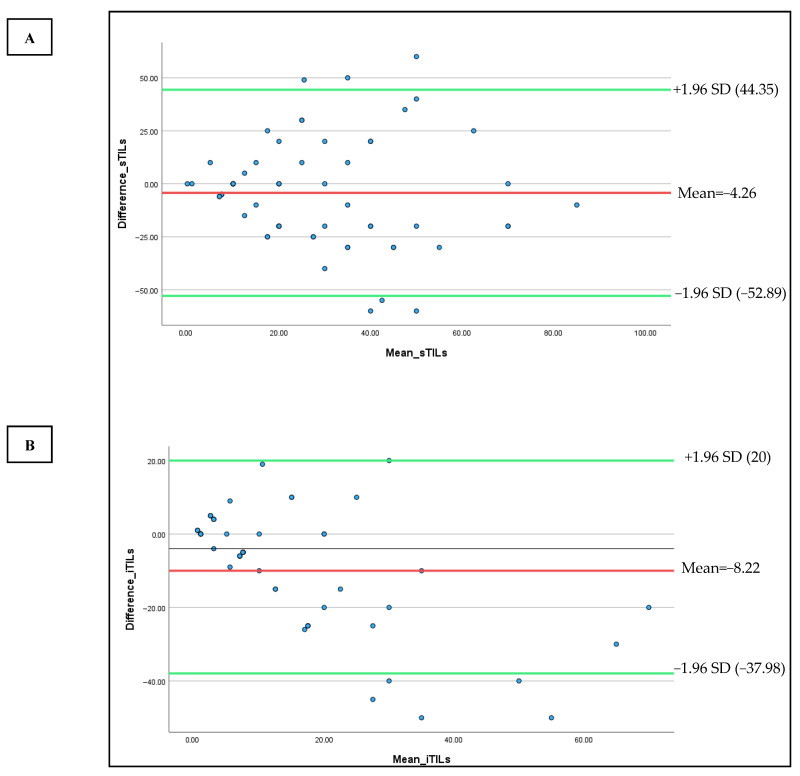
Bland–Altman plots were used to compare the sTILs (**A**) and iTILs (**B**) values of two observers (P1 and P2) to the median of their scores. The Y-axis represents the difference between the observer and the X-axis shows their mean values. The red line denotes the mean difference, and the green lines show the upper and lower limits of agreement.

**Figure 3 diagnostics-15-02492-f003:**
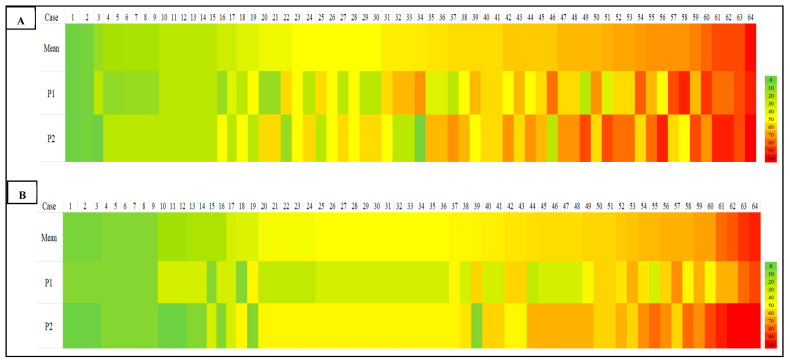
Heat map for scoring of sTILs (**A**) and iTILs (**B**) by two pathologists (P1 and P2). The first row shows the mean value for each case scored by both pathologists in an ascending manner from left to right; the second and third rows represent two observers’ sTIL and iTIL records for the similar case. Color-coding indicates the percentage density of TILs quantified by each pathologist, from green (0%) to yellow (50%) and red (100%).

**Table 1 diagnostics-15-02492-t001:** Different cut-off systems for TIL assessment.

Cut-Off System	TILs Classification	
	Low TILs	High TILs
Cut-off value 1	≤10%	>10%
Cut-off value 2	≤20%	>20%
Cut-off value 3	≤30%	>30%
Cut-off value 4	≤40%	>40%
Cut-off value 5	≤50%	>50%

**Table 2 diagnostics-15-02492-t002:** Intraclass correlation coefficients for consistency in sTIL and iTIL assessment.

Reliability Measure	sTILs ICC (95% CI)	iTILs ICC (95% CI)
ICC (agreement)	0.57	0.70
ICC (consistency)	0.58	0.75

Abbreviations: sTILs, stromal tumor-infiltrating lymphocytes; iTILs, intratumoral tumor-infiltrating lymphocytes; ICC, intraclass correlation coefficient.

**Table 3 diagnostics-15-02492-t003:** Cohen’s kappa (κ) coefficient for interobserver agreement in sTILs and iTILs at various cut-off thresholds between P1 and P2.

Cut-Off (%)	sTILs (κ)	iTILs (κ)
10	0.40 *	0.43 *
20	0.29	0.31
30	0.13	0.25
40	0.16	0.48 *
50	0.34	0.38

Note: Asterisks (*) indicate the highest agreement values for each category.

**Table 4 diagnostics-15-02492-t004:** Intraclass correlation coefficients for interobserver agreement in discrepant cases (P1, P2, and P3).

Reliability Measure	sTILs ICC (95% CI)	iTILs ICC (95% CI)
ICC (agreement)	0.70	0.81
ICC (consistency)	0.81	0.84

Abbreviations: sTILs, stromal tumor-infiltrating lymphocytes; iTILs, intratumoral tumor-infiltrating lymphocytes; ICC, intraclass correlation coefficient; CI, confidence interval.

**Table 5 diagnostics-15-02492-t005:** Pairwise Cohen’s kappa (κ) coefficient values for sTILs and iTILs at different cut-off thresholds among P1, P2, and P3.

Cut-Off (%)	sTILs			iTILs			Interpretation
	P1 vs. P3 (κ)	P2 vs. P3 (κ)	Average (κ)	P1 vs. P3 (κ)	P2 vs. P3 (κ)	Average (κ)	
10	0.50	0.36	0.43	0.48	0.50	0.49	Fair agreement
20	0.44	0.35	0.40	0.62	0.27	0.45	Fair agreement
30	0.50	0.25	0.40	0.64	0.47	0.56	Fair agreement
40	0.50	0.50	0.50	0.65	0.64	0.65	Fair agreement
50	0.65	0.68	0.67 *	1.0	0.48	0.74 *	Substantial agreement

Abbreviations: κ, Cohen’s kappa coefficient. Note: Asterisks (*) indicate the highest average agreement values for each category.

**Table 6 diagnostics-15-02492-t006:** Frequency of morphologic features contributing to interobserver discrepancies (*n* = 36).

Morphologic Features	Number of Cases (*n*)	Percentage (%)
Heterogeneous distribution of TILs	26	32
Poorly defined tumor–stroma interface	24	30
Focal dense lymphoid aggregates	16	20
Immune cell mimics/Reactive plasma cells mimicking tumor cells	13	15
Extensive tumor necrosis	2	3

Note: most cases exhibited more than one contributing feature.

## Data Availability

Upon request from the corresponding author on reasonable request.

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
