# Peer review of "Assessment of Tumor-Infiltrating Lymphocytes in Triple-Negative Breast Cancer: Interobserver Variability and Contributing Factors"

_diagnostics, 2025, doi:10.3390/diagnostics15192492_

Round 1

Reviewer 1 Report

Comments and Suggestions for Authors

Specific comments for the authors:

In their manuscript entitled Bridging the Gap: Assessing Interobserver Variability in Tumour-Infiltrating Lymphocyte (TIL) Scoring for Triple-Negative Breast Cancer, the authors examined interobserver agreement among pathologists when scoring stromal and intratumoral TILs on H&E-stained triple-negative breast cancer (TNBC) slides. They also identified histological factors that contribute to this variability. They assessed TILs in 64 TNBC cases according to Working Group (WG) guidelines and calculated the intraclass correlation coefficient (ICC) and Cohen's kappa coefficient to quantify interobserver agreement.

Based on their investigations, the authors demonstrated that:

(i) there was moderate interobserver agreement for stromal TILs (ICC = 0.58) and strong agreement for intratumoral TILs (ICC = 0.71);

(ii) this was significantly related to three main confounding variables: heterogeneous TIL distribution, a poorly-defined tumour-stroma interface, and focal dense lymphoid infiltrates. In conclusion, the authors emphasised the urgent need for standardised TIL assessment criteria to reduce interobserver variability and improve evaluation reproducibility. The manuscript is mostly comprehensible and convincing. The methods are well described. Although the results and discussion are clearly presented, the authors could make some major changes to improve the manuscript (see specific comments).

In conclusion, the presented data are of interest in some respects. After incorporating the specific comments mentioned below, the manuscript has the potential to be accepted.

Major comments:

The investigations reveal more questions than answers regarding the critical issue of observers' independent assessment of TILs, particularly in circumstances involving therapeutic options. There is definitely a lack of a “gold standard” for quantifying TILs, which the authors must address. Furthermore, the possible factors influencing observer inconsistency must be investigated in more detail.

Specific comments:

Title: As the title implies, the results showed only highly variable TIL scoring between observers. Therefore, the title should be adapted accordingly.

Abstract: In the conclusion, the authors state that an AI-based scoring method could help to overcome interobserver variability, even though AI-based scoring methods were not part of the investigation.

Material and Methods: Please add more information or data on the learning curve relating to the following sentence: 'A calibration session was conducted prior to scoring to ensure adherence to the protocol and minimise interpretive variability.' Regarding the sentence 'The annotation and scoring process were guided by corresponding IHC, CD4+ and CD8+ markers to enhance the accuracy of lymphocyte identification and to minimise ambiguity in distinguishing TILs within the tumour microenvironment', immunohistochemical application of CD3 should be used to quantify the total number of T cells. Regarding Table 1, TIL quantification should be performed using automated methods (e.g. ImageJ/FIJI plugins or QuPath) to establish a gold standard.

Results:

# Tables 3 to 5, as well as Figures 2 and 3, show that the Cohen's kappa values, the Bland–Altman plots and the heat map for scoring display high variability, low robustness and high randomness, even in the case of the learning sessions performed, which is not comprehensible overall. The authors must mention and discuss this.

# The sentence ‘Consensus review methodologies have proven valuable in diagnostic settings involving subjective assessments and may serve as a quality control measure in pathology workflows, especially in multicentre trials or AI training datasets (39). These improvements underscore the importance of incorporating consensus-based scoring into clinical and research workflows, particularly when addressing cases that exhibit significant interobserver variability' is an interpretation and not a result. Please re-write adequately.

# 4.6. Contributing factors to interobserver discrepancies: This chapter is largely descriptive and must be quantified using dichotomous parameters for heterogeneous TIL distribution (yes/no), poorly-defined tumour-stroma interfaces (yes/no) and focal dense lymphoid infiltrates (yes/no).

Conclusion: The conclusion is overly optimistic in relation to the data presented on observer agreement overall. Therefore, the authors should make substantial recommendations on how TIL quantification could be performed consistently in future.

Author Response

Dear Reviewer,

Thank you for your valuable comments and suggestions. We have carefully addressed each of your points in detail, and the responses are elaborated in the attached document titled “Response to Reviewer 1” The document provides a point-by-point reply to all comments, along with clear indications of the revisions made in the manuscript (including page numbers, paragraph locations, and updated text where applicable).

We hope the revised manuscript and detailed responses adequately address your concerns. Please do not hesitate to let us know if further clarification is required.

Sincerely,
Nurkhairul Bariyah
On behalf of all co-authors

Reviewer 2 Report

Comments and Suggestions for Authors

The authors examined interobserver variability of tumor-infiltrating lymphocytes (TILs) in triple-negative breast cancer (TNBC). Although some data is interesting, there must be several points to be addressed.

Overall, the manuscript lacks novelty. Interobserver variability of assessing TILs has been previously reported by several researchers. The authors should make it clear what is new in the present study.

Distribution of TIL in each observe is not given. How was it different between two pathologists? While the authors described that agreement declined with increasing cut-off thresholds (Table 3), it seems that kappa values were relatively high when both lowest (10%) and highest (50%) cut-off were used, while intermediate value (30%) made lower kappa value.

It is obvious that kappa value was low if the threshold is set to a value close to the average.

 Clinicopathological background affecting TIL assessment should be considered (Tumor size, status of neoadjuvant chemotherapy, etc).

Regarding Table 2 and Table 4, 95% CI is not given.

Please explain in more detail about ICC (agreement) and ICC (consistency). How are they different?

Author Response

Dear Reviewer,

Thank you for your valuable comments and suggestions. We have carefully addressed each of your points in detail, and the responses are elaborated in the attached document titled “Response to Reviewer 2.” The document provides a point-by-point reply to all comments, along with clear indications of the revisions made in the manuscript (including page numbers, paragraph locations, and updated text where applicable).

We hope the revised manuscript and detailed responses adequately address your concerns. Please do not hesitate to let us know if further clarification is required.

Sincerely,
Nurkhairul Bariyah
On behalf of all co-authors

Round 2

Reviewer 2 Report

Comments and Suggestions for Authors

The manuscript is well revised and suitable for publication.